# Interaction Effects of Cultivars and Nutrition on Quality and Yield of Tomato

Oana-Raluca Rusu [1], Ionel Mangalagiu [2], Dorina Amăriucăi-Mantu [2], Gabriel-Ciprian Teliban [3], Alexandru Cojocaru [3], Marian Burducea [3,4], Gabriela Mihalache [5], Mihaela Roșca [3], Gianluca Caruso [6,*], Agnieszka Sekara [7] and Vasile Stoleru [3,*]

[1] Department of Public Health, "Ion Ionescu de la Brad" University of Life Sciences, 6 M. Sadoveanu, 700449 Iasi, Romania; raluca.rusu@uaiasi.ro

[2] Department of Chemistry, "Alexandru Ioan Cuza" University of Iasi, Carol I Blvd. 20A, 700506 Iasi, Romania; ionelm@uaic.ro (I.M.); dorinaiasi@yahoo.com (D.A.-M.)

[3] Department of Horticulture, "Ion Ionescu de la Brad" University of Life Sciences, 3 M. Sadoveanu, 700440 Iasi, Romania; gabrielteliban@uaiasi.ro (G.-C.T.); acojocaru@uaiasi.ro (A.C.); mihaelarosca@uaiasi.ro (M.R.)

[4] Research and Development Station for Aquaculture and Aquatic Ecology, "Alexandru Ioan Cuza" University, Carol I Blvd, 700506 Iasi, Romania

[5] Integrated Center of Environmental Science Studies in the North East Region, "Alexandru Ioan Cuza" University, Carol I Blvd, 700506 Iasi, Romania

[6] Department of Agricultural Sciences, University of Naples Federico II, Portici, 80055 Naples, Italy

[7] Department of Horticulture, Faculty of Biotechnology and Horticulture, University of Agriculture, 31-120 Krakow, Poland; agnieszka.sekara@urk.edu.pl

* Correspondence: gcaruso@unina.it (G.C.); vstoleru@uaiasi.ro (V.S.); Tel.: +39-320-763-083 (G.C.); +40-743-180-275 (V.S.)

**Abstract:** Tomato is considered the most important vegetable crop worldwide. Improving the nutritional value of fruits must be based on sustainable production in terms of varieties and fertilization management. This study aimed to improve the nutritional value (total soluble solids, acidity, lycopene, β-carotene, polyphenols, macro and microelements) of two tomato varieties ('Cristal' and 'Siriana') under three fertilization types (NPK chemical fertilizer, chicken manure and biological fertilizer with microorganisms) for the greenhouse. Primary metabolism compounds do not vary significantly according to the type of fertilizer used. The results for the antioxidant compounds showed a better effect of biological fertilization compared to chemical fertilizer and control unfertilized. Thus, the antioxidant activity was improved by 28% compared to chemical fertilization, the lycopene content by 36% and β-carotene by 96%, respectively. The tomato fruits from the local cultivar ('Siriana') are richer in nutritional compounds such as rutin, regardless of the type of fertilization, which denotes a good ability to adapt to crop conditions. Tomato cultivars reacted positively to microbiological fertilization compared to chemical, thus producing nutritious fruits under sustainable management. Tomato fruits were richer in the quality of microelement contents.

**Keywords:** *Solanum lycopersicum* L.; chemical; organic and biological fertilization; antioxidants; minerals

## 1. Introduction

Vegetable consumption provides vitamins, minerals, dietary fibers and bioactive compounds to the human body. In addition, vegetables are low in calories and have been strongly associated with the prevention of various chronic diseases. Adequate consumption of vegetables also prevents micronutrient deficiencies in the human body, especially in less developed countries [1–3]. Moreover, a low intake of fruit and vegetable is among the top ten risk factors for global mortality [4]. Thus, for a healthy life, the World Health Organization has recommended the daily consumption of at least 400 g of fruit and vegetables [5]. So, in the last decade, due to the increasing number of nutritional diseases,



there is a growing interest to enhance the antioxidant compound input in diets to prevent premature, unhealthy aging. Food diets based on fruits and vegetables rich in polyphenols are closely related to the reduction of neurodegenerative and cardiovascular diseases [6], and related to age and cancer progression [7–9]. The consumption of vegetables rich in flavonoids is linked to beneficial health outcomes [10,11]. Strong evidence suggests that flavonoids may be able to reduce oxidative stress and damage-associated problems related to these diseases. Consequently, farmers and consumers are increasingly interested in obtaining fresh vegetables and fruit with a high flavonoid content [12]. Phenolic compounds, such as rutoside, *p*-coumaric acid, and quercitrin, are known for their favorable impact on the human body, and the high nutritional health of the population [6,13].

Tomato is among the most consumed vegetable worldwide, being produced in 2020, approximately 186.82 million metric tons [14]. Tomato fruits; rich in all macro- and micronutrients bring a balanced elemental composition that is essential for the synthesis of primary and secondary metabolites, which also improves the organoleptic and biochemical qualities of the fruits. The secondary metabolites are remarkably present in Solanaceae species, i.e., phenolic compounds (phenolic acids and flavonoids), carotenoids (lycopene, carotene), vitamins (ascorbic acid and vitamin A) and glycoalkaloids [15,16]. Some of the mentioned substances, especially lycopene, have high antioxidant activity which is important for the prevention of cardiovascular and oncological diseases [17,18]. According to Rosa-Martínez et al. [17] 200 g of fresh tomato provides 30 to 36% of the Recommended Dietary Allowance (RDA) for vitamin C, 10% of the Adequate Intake (AI) for K, and 5–10% of the RDA for P and Mg. A portion of 100 g of tomato provides the daily required intake of vitamin C and tocopherol, and provides 5–10% of provitamin A intake.

Due to the high content in lycopene and other bioactive substances, it is recommended that tomatoes be included in the daily diet of consumers [7,18,19]. Therefore, improving their nutrient content could lead to significant health benefits. This can be achieved, either by using valuable genotypes that have adapted to local conditions, or by optimizing the plant nutrition regime [20]. For instance, the content of vitamins (Vit. A, C, K, vitamin B complex) and minerals (Fe, Mg, Zn, K, Ca, P, Mn, Cu) depends on variety, cultivation system, fertilizer type, and harvest time [10,21].

Fertilizers play the main role in enhancing the nutritional status of plants. Chemical fertilization is a widely used method for increasing crop yields, but it has been found to have significant negative impacts on the environment and human health. For instance, chemical fertilizers are one of the main contributors to the greenhouse effect or soil salinization. Organic fertilization represents an alternative that has the potential to reduce the negative effects of chemical fertilization [22]. Organic fertilizers can be of plant, animal or microorganism origin. Among organic fertilizers are composts (e.g., vermicompost, water hyacinth compost, village or town compost), farmyard manure (e.g., cattle and poultry manures), green manures (leguminous and non-leguminous plants) or biofertilizers (algal, fungal and bacterial). The advantages of using organic fertilizers refer to better physical and physiological structure of the soil, enhanced biological activity, slow release of nutrients, improvement of organic matter, and reduced loss of nutrients. In addition to the benefits related to the environment, organic fertilization has also a positive impact on the quality of vegetable products and harvest quantity [23]. For instance, in tomatoes, it was shown that organic or biological farming had a favorable impact on the improvement of polyphenolic compounds and the antioxidant capacity of fresh fruits or processed vegetables [24].

Research conducted by Rosa-Martínez et al. [17] and Martí et al. [25] has shown that choosing the proper cultivar and fertilizer technology provides the possibility to produce tomato fruits with improved bioactive compounds in an open field under different climatic conditions. As Assefa and Tadesse [26] explained, the nutrients were released more slowly to the plants by organic fertilizer than chemical fertilizers; thus, allowing the plants to process the fertilizer in a more natural way and avoid the excessive fertilization that could be harmful to them.

The use of chemical fertilizers in intensive agricultural practices can have negative environmental and health impacts, leading to increased interest in organic and biological fertilizers. Furthermore, the Circular Economy Action Plan adopted by European Union promote the recycling of nutrients from manure and other organic sources to replace chemical fertilizers [27]. Starting in 2023, the Common Agricultural Policy (CAP) aims "to put agriculture closer in line with the targets of the Farm to Fork strategy with regard to reduced nutrient pollution" [28]. In Romania, under the Common Agricultural Policy, six eco-schemes have been envisaged for 2023–2027. One of them is the conversion to organic farming [29]. Therefore, in this study, we investigated the effects of different fertilization regimes on the tomato fruits quality and nutrient content of two cultivars of tomato ('Cristal' and 'Siriana') that are known for their adaptability to the environmental conditions of Romania, resistance to pests, high productivity and fruit quality.

While previous studies have explored the effects of different fertilization regimes on plant growth and fruit quality, the novelty of our work lies in the use of specific fertilizers and the evaluation of a wider range of parameters, including micro and macronutrient content, total polyphenol content, lycopene, β-carotene, antioxidant activity and production. Specifically, we tested the effects of chemical fertilizers (NPK 20-20-20), organic fertilizers (chicken manure) and biological fertilizers on the fruit quality and quantity.

## 2. Materials and Methods

### 2.1. Experimental Site and Growing Conditions

The research was carried out with two tomato varieties ('Cristal' and 'Siriana') in the greenhouse at the experimental farm of Iasi University of Life Sciences, Romania (N = 47°11′76″ E = 27°33′71″), during 2019 and 2020, from mid-April to the end of October. Varieties selection was conducted based on their adaptability to the environmental conditions of Romania, their resistance to pests and their higher net return advantages. 'Siriana' is a cultivar that was obtained at the Vegetable Research and Development Station in Buzau, and is one of the most cultivated in Romania. This cultivar produces its first fruits after ~100 days of germination, with a medium weight of the fruit of 140 g and a production of 5–5.5 kg of fruits per plant [10]. 'Cristal' is an international and early tomato cultivar with indeterminate growth and intended for cultivation in greenhouses [30].

Seedlings that were 55 days old, were grown in multicell trays in compliance with the organic regulation [5,20], and transplanted to a greenhouse during mid-April at a density of 2.5 plants·m$^{-1}$. The soil from the greenhouse that was used for the experiment [10,31] is characterized as a chernozem loam-clay with pH 7.20; electrical conductivity (EC) 478 μS·cm$^{-2}$, CaCO$_3$ 0.41%, organic matter (OM) 28.56 mg·kg$^{-1}$, 5.91 C/N, 2.8 g·kg$^{-1}$ N, 32 mg·kg$^{-1}$ available P, 218 mg·kg$^{-1}$ available K. The temperature, relative humidity and light intensity recorded in the research years are reported in Figure 1.

### 2.2. Experimental Design, Plants and Treatments

The experimental protocol was based on the factorial combination between 2 cultivars and 3 fertilization types, plus an untreated control using a split-plot design with 3 replicates with 18 plants per plot, and with a surface of 7.2 square meters for each plot. Varieties were represented by two cultivars of tomato, 'Siriana' and 'Cristal'. The second factor was represented by three fertilization types: (I) chemical fertilization, (II) organic fertilization, and (III) biological fertilization, compared with the unfertilized version.

The NPK chemical fertilization was performed with 200 kg·ha$^{-1}$ of Nutrifine® NPK 20-20-20, applied before transplant, and 2 supplies of 300 kg·ha$^{-1}$ of Nutrifine® NPK 9-18-27 + 2 MgO. The first application was carried out 3 weeks after planting, and the second fertilization was carried out 3 weeks after the first application. Organic fertilization represented by chicken manure was applied by 2000 kg·ha$^{-1}$ of commercial Orgevit® in two phases: 1250 kg·ha$^{-1}$ before transplant, and the remainder 30 days after planting. Orgevit® is a granular fertilizer with 65% OM, pH 7, 4% N, 3% P$_2$O$_5$, 2.5% K$_2$O, 1% MgO, 0.02% Fe, 0.01% Mn, 0.01% B, 0.01% Zn, 0.001% Cu, 0.001% Mo.

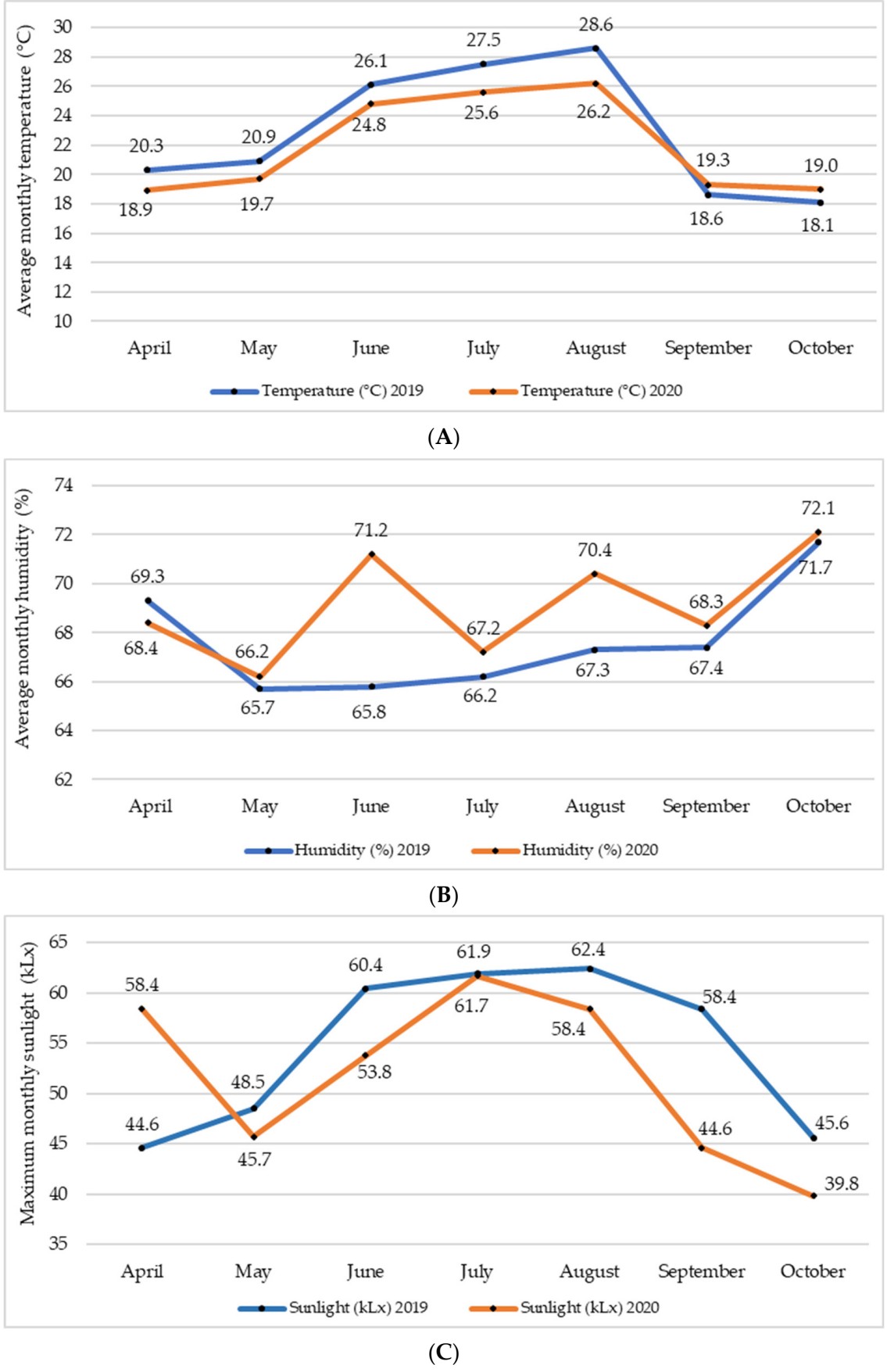

**Figure 1.** Climatic conditions during the experiment (2019–2020): (**A**) average monthly temperature; (**B**) average monthly humidity; (**C**) maximum monthly sunlight.

The same amount of active substance from the two fertilizers (chemical and organic used in the first year) was used, taking into account that the plants use approximately 70% of the organic fertilizer in the first year.

The biological fertilization consisted of the application of Micoseed® at 30 kg·ha⁻¹ split in two equal doses that were supplied before transplant and 30 days after planting, integrated with 5 L·ha⁻¹ Nutryaction® according with the manufacturer's recommendations. The biological product is based on microorganisms that predominantly contain arbuscular mycorrhizal fungi spores of *Claroideoglomus etunicatum*, *Funneliformis mosseae*, *Glomus aggregatum*, *Rhizophagus intraradices*. In addition, the product is complexed with fungi and bacteria species belonging to the genera Trichoderma, Streptomyces, Bacillus, Pseudomonas.

During the crop cycles the following practices were performed: drip irrigation; stringing; trimming with branches; and old leaves removal [31]. The control of diseases and pests was carried out with products allowed by the organic regulation.

### 2.3. Total Soluble Solids, Acidity and Ash Analyses

The total soluble solids (TSS) of tomato juice were measured with a digital refractometer (RX5000α, Atago, Tokyo, Japan). The acidity, in terms of the content of citric acid, was quantified by titration with a titrator (808 Titrando, Metrohm, Wesbury, NY, USA) using 6.0 g of tomato juice, which was diluted with 60 mL of ultra-clean DI water [32].

The ash content and the calorific value were determined according to AOAC, 2005 [33].

### 2.4. Analytical Quantifications of the Phenolic Compounds

Twenty fruit-random samples, individually weighing 2500–3000 g were collected in each plot at commercial maturity (BBCH 803–805) for laboratory analyses. The samples used for the analyses of antioxidants (polyphenol contents, lycopene and β–carotene) were prepared as follows: fruits were cut into 1 cm fragments and dried on a Sanyo stove, type MOV-112F, at a temperature of 70 °C until constant weight. The samples were then ground into small fragments of 0.1–1 mm.

For the analysis of the phenolic compounds, an aqueous extraction was performed at 38 °C under shaking for 1 h. After filtration, the extracts were kept in a freezer at −24 °C prior to HPLC and spectrophotometric analyses. The total phenolic compounds were determined by the Folin–Ciocâlteu method based on the compounds' property to reduce sodium phosphate-wolframite in an alkaline medium, to the blue oxide of wolfram [34]. According to Liu et al. [35], 2 mL of extract, 10 mL of Folin–Ciocâlteu reagent (1:10) and 8 mL $Na_2CO_3$ solution of 75% were added to modify the experimental conditions. The mixture was allowed to stay at room temperature for 120 min and afterward, the absorbance was measured using a spectrophotometer Jasco V530, at 750 nm and compared to a gallic acid (GA) calibration curve. The results were expressed as mg gallic acid equivalent (GAE) per ml (μg·100 mL⁻¹).

The individual phenolic compounds were determined using RP-HPLC with a UV detector coupled with MS. The detection and quantification of polyphenols was conducted in triplicate. A number of 17 standard solutions were used to carry out the analyses: caffeic acid, chlorogenic acid, *p*-coumaric acid, ferulic acid, gentisic acid, sinapic acid, caftaric acid, kaempferol, apigenin, rutin, quercetin, quercetin3-β-D-glucoside, hyperoside, myricetin, isoquercitrin, fisetinpatuletin, and luteolin.

HPLC analysis was performed with an Agilent 1100 Series HPLC system equipped with a degasser, binary pump, and an autosampler, and a reversed-phase column Zorbax SB-C18 (100 mm × 3.0 mm, 3.5 μm) with an operating temperature of 48 °C. For the mobile phase, the methanol procedure was: 0.1% acetic acid, with flow rate 1 mL/min, and an injection volume of 5 μL; and the gradient of elution was 5% to 42% methanol for 35 min, and the isocratic elution was performed with 42% methanol for 3 min.

UV detection was performed at 330 nm during the first 17.5 min and then at 370 nm for up to 35 min, using a DAD detector.

For the MS detection, we used a mass spectrometer: Agilent 1100 MSD ion-trap interface Turbo ion spray (ESI) in a negative ionization mode, nitrogen gas with flow rate 12 L/min, ionization source temperature at 360 °C, and nebulizer; nitrogen at 70 psi with capillary voltage 3000. The analysis was multiple reaction-type monitoring (MRM) and single ion monitoring (SIM). Full identification of compounds was performed both by UV and MS detection.

The quantitative determination was performed by UV through an external standard method using a calibration curve with good linearity in the range of 0.5–50 $\mu g \cdot mL^{-1}$ [36,37]. The calibration curve showed good linearity within the mentioned range and the regression coefficient was $R^2 = 0.9943$, which shows that there is a positive correlation between the polyphenolic content of *tomato* fruits and the applied fertilization system.

### 2.5. Lycopene and β-Carotene Content

The lycopene content was determined according to the spectrophotometric method described by Davis et al. [38], while the utilized method for determining β-carotene in tomatoes was described by Cadoni et al. [39]. Spectrophotometric readings were performed at 452 nm for β–carotene and 472 nm for lycopene. The extracts were prepared using fresh biomass.

### 2.6. Antioxidant Activity

The neutralizing effect of the DPPH free radical was calculated at three different concentrations of ethanol extracts: 0.05 mL of 10, 5, and 2.5 $mg \cdot mL-1$ extracts were mixed with 2.95 mL solution DPPH. After a 5 min reaction time, the absorption was measured at 420 nm using methanol as a blank. Approximately 10 g of tomatoes was extracted in 100 mL of 80% aqueous solution of ethanol at room temperature for 1 h. The extracts were filtered and the filters were left to evaporate in a dry environment. The percentage of free-radical scavenging activity was determined using the following formula: $100 \times (Ai - Af)/Ai$, where Ai = the absorption before the addition of the tested extract, and Af = the absorption value after 5 min reaction time. Trolox (6-hydroxy$-$2,5,7,8-tetramethylchroman-2-carboxylic acid) was used as a positive control. At a concentration of 2.5 mM Trolox was able to fully neutralize the level of DPPH radical used. The Trolox equivalent antioxidant capacity (TEAC) of the extracts was calculated by comparing their percentage of the free-radical scavenging activity of DPPH with the Trolox standard curve [40].

### 2.7. Mineral Content

The mineral concentration (macro- and microelements) of the tomato fruits was determined by using the atomic-absorption spectrometry method. The fruits were first oven-dried at 105 °C for 48 h, and then samples of 0.5 and 1 g were digested [41]. The minerals were extracted with nitric acid ($HNO_3$) and hydrochloric acid (HCl) 1:1 following the procedure described by Fernández-Ruiz et al. [41]. The solutions obtained were analyzed by the atomic absorption spectrometer Contra 300 (Analytik Jena, Göettingen, Germany).

### 2.8. Biometric and Yield Determinations

For each plot and harvest, 20- ripe fruits were determined for the mean fruit weight. The yield ($kg \cdot ha^{-1}$) was calculated by using the following formula [10]: Yield ($kg \cdot ha^{-1}$) = (plants/ha × fruits/plant × average fruit weight)/1000.

### 2.9. Statistical Analysis of the Data

The results are expressed as means ± SD of the two experimental years. The data were statistically processed by one-way ANOVA. The significant differences between treatments were established by using Duncan's test with a degree of confidence of 95% ($p \leq 0.05$), using SPSS software version 21 (IBM Microsoft, New York, NY, USA).

## 3. Results

### 3.1. Water Content, Dry Matter, Total Soluble Solid, Acidity and Ash

The effect of the tomato cultivar and the fertilization regime on the water, dry matter, TSS, acidity, and ash content is shown in Table 1. Data presented for both technological factors did not induce significant differences with reference to the mentioned parameters (Table 1). However, in the case of TSS, the 'Siriana' cultivar determined higher values of the cu parameter with 8.8%. Additionally, fertilization with Micoseed determined higher values of TSS; 18.03% as compared to the control.

**Table 1.** Influence of tomato cultivar and fertilization type on water, dry matter, total soluble solids, acidity and ash content.

| Treatment | Water Content (%) | Dry Matter (%) | TSS (°Brix) | Acidity (g Citric Acid·100 g$^{-1}$ f.w.) | Ash (g·100 g$^{-1}$ d.w.) |
|---|---|---|---|---|---|
| | | | Cultivar | | |
| Cristal | 94.03 ± 3.14 | 5.97 ± 3.14 | 5.78 ± 1.07 | 0.50 ± 0.11 | 4.57 ± 0.84 |
| Siriana | 93.59 ± 3.03 | 6.41 ± 3.03 | 6.29 ± 1.08 | 0.55 ± 0.11 | 5.01 ± 0.84 |
| Signification | ns | ns | ns | ns | ns |
| | | | Fertilization type | | |
| Nutrifine | 94.01 ± 3.51 | 5.99 ± 3.51 | 5.99 ± 1.04 | 0.50 ± 0.09 | 4.51 ± 0.85 |
| Orgevit | 93.53 ± 3.33 | 6.47 ± 3.33 | 6.18 ± 1.08 | 0.56 ± 0.10 | 5.12 ± 0.89 |
| Micoseed | 93.69 ± 3.10 | 6.32 ± 3.10 | 6.48 ± 1.16 | 0.60 ± 0.10 | 4.92 ± 0.86 |
| Control | 94.02 ± 2.98 | 5.98 ± 2.98 | 5.49 ± 1.09 | 0.43 ± 0.09 | 4.60 ± 0.89 |
| Signification | ns | ns | ns | ns | ns |

Within each column: ns—no statistically significant difference for $p < 0.05$ according to Duncan's test.

The effect of the interactions between tomato cultivars and the fertilization type on the water content, dry matter, TSS, acidity and ash are shown in Table 2. No significant differences were recorded for the water content, dry matter, TSS and ash regardless of the cultivar and fertilization type. Only for acidity were significant differences recorded between the fruits of the unfertilized 'Cristal' tomato cultivar (0.4 g citric acid·100 g$^{-1}$ f.w.), which registered the lowest value, and the fruits of 'Siriana' biologically fertilized (0.61 g citric acid·100 g$^{-1}$ f.w.), which provided the highest value of the citric acid.

**Table 2.** Water, dry matter, total soluble solid, acidity and ash content in tomato fruits.

| Treatment | Water Content (%) | Dry Matter (%) | TSS (°Brix) | Acidity (g Citric Acid·100 g$^{-1}$ f.w.) | Ash (g·100 g$^{-1}$ d.w.) |
|---|---|---|---|---|---|
| Cristal × Nutrifine | 94.30 ± 2.53 | 5.70 ± 1.10 | 5.88 ± 1.13 | 0.48 ± 0.09ab | 4.22 ± 0.20 |
| Cristal × Orgevit | 93.61 ± 1.87 | 6.39 ± 1.24 | 6.02 ± 1.16 | 0.53 ± 0.10ab | 5.04 ± 0.98 |
| Cristal × Micoseed | 93.83 ± 2.09 | 6.17 ± 1.19 | 6.21 ± 1.20 | 0.59 ± 0.11ab | 4.77 ± 0.92 |
| Cristal × Control | 94.39 ± 1.89 | 5.61 ± 1.08 | 5.01 ± 0.97 | 0.40 ± 0.08b | 4.23 ± 0.82 |
| Siriana × Nutrifine | 93.72 ± 1.94 | 6.28 ± 1.21 | 6.11 ± 1.18 | 0.52 ± 0.10ab | 4.80 ± 0.93 |
| Siriana × Orgevit | 93.45 ± 2.39 | 6.55 ± 1.27 | 6.34 ± 1.22 | 0.58 ± 0.11ab | 5.20 ± 1.00 |
| Siriana × Micoseed | 93.54 ± 1.90 | 6.46 ± 1.25 | 6.75 ± 1.31 | 0.61 ± 0.12a | 5.06 ± 0.98 |
| Siriana × Control | 93.65 ± 1.92 | 6.35 ± 1.22 | 5.97 ± 1.15 | 0.47 ± 0.09ab | 4.97 ± 0.96 |
| Signification | ns | ns | ns | * | ns |

Within each column: ns—no statistically significant difference; *—significant differences; values associated with the same lower-case letters are not statistically different at $p \leq 0.05$ according to Duncan's test.

### 3.2. Antioxidant Contents

The effects of the tomato cultivar and the fertilization regime on total phenolic, lycopene, β-carotene content and antioxidant activity are shown in Table 3. Data for variety factor highlights significant differences in phenolic content and antioxidant activity. However, native cultivar, represented by 'Siriana' obtained higher results for mentioned

parameters, which indicates that it can be promoted with good results regarding the activity of antioxidant compounds.

**Table 3.** Influence of tomato cultivar and fertilization type on total phenols, lycopene, β-carotene and antioxidant activity.

| Treatment | Total Phenolic ($\mu g \cdot 100$ mL$^{-1}$) | Lycopene (mg·100 g$^{-1}$ d.w.) | β-Carotene (mg 100 g$^{-1}$ d.w.) | Antioxidant Activity (mmol Trol·100 g$^{-1}$ d.w.) |
|---|---|---|---|---|
| | | Cultivar | | |
| Cristal | $194.79 \pm 42.66$ | $9.01 \pm 1.87$ | $3.04 \pm 0.93$ | $85.88 \pm 17.45$ |
| Siriana | $274.52 \pm 57.86$ | $10.15 \pm 2.20$ | $3.43 \pm 1.10$ | $109.56 \pm 21.22$ |
| Signification | * | ns | ns | * |
| | | Fertilization type | | |
| Nutrifine | $195.12 \pm 58.07$ | $8.45 \pm 1.51$ b | $2.28 \pm 0.41$ c | $83.53 \pm 21.48$ |
| Orgevit | $251.82 \pm 71.05$ | $9.67 \pm 1.71$ ab | $3.39 \pm 0.60$ b | $104.26 \pm 23.06$ |
| Micoseed | $270.12 \pm 67.61$ | $11.51 \pm 2.20$ a | $4.48 \pm 0.86$ a | $107.56 \pm 24.95$ |
| Control | $221.58 \pm 45.43$ | $8.70 \pm 1.71$ b | $2.79 \pm 0.54$ bc | $95.54 \pm 17.41$ |
| Signification | ns | * | * | ns |

Within each column: ns—no statistically significant difference; *—significant differences; values associated with the same lowercase letters are not statistically different at $p < 0.05$ according to Duncan's test.

The fertilization factor induces significant values for the lycopene and β-carotene compounds. Thus, fertilization with Micoseed obtained higher values by 36% in the case of lycopene and 96% in the case of β-carotene compared to Nutrifine, which means that the fertilization is not appropriate or the dose of fertilizer can be improved.

The total content of polyphenols, lycopene, β-carotene and antioxidant activity is influenced by interactions between cultivars and fertilization type. The content of total polyphenols showed a wide range according to the cultivar and fertilization type. In the tomato, it varied from 152.47 $\mu g \cdot 100$ mL$^{-1}$ in 'Cristal' cv. under chemical fertilization to 314.23 $\mu g \cdot 100$ mL$^{-1}$ in the 'Siriana' cv. microbiological fertilized. The highest value in 'Siriana' was also recorded under organic fertilization (Table 4). The lycopene content in tomato increased with 55% from 'Siriana' cv., biologically fertilized than control, and antioxidant activity was improved with 77% than 'Cristal' chemically fertilized.

**Table 4.** Total polyphenol content, lycopene, β-carotene and antioxidant activity in tomato fruits.

| Treatment | Total Phenolic ($\mu g \cdot 100$ mL$^{-1}$) | Lycopene (mg·100 g$^{-1}$ f.w.) | β-Carotene (mg 100 g$^{-1}$ f.w.) | Antioxidant Activity (mmol Trolox·100 g$^{-1}$ d.w.) |
|---|---|---|---|---|
| Cristal × Nutrifine | $152.47 \pm 29.45$ c | $8.11 \pm 1.56$ b | $2.19 \pm 0.42$ d | $69.18 \pm 13.36$ b |
| Cristal × Orgevit | $201.17 \pm 38.86$ bc | $9.32 \pm 1.80$ ab | $3.26 \pm 0.63$ bcd | $91.27 \pm 17.63$ ab |
| Cristal × Micoseed | $226.00 \pm 43.65$ abc | $10.65 \pm 2.06$ ab | $4.15 \pm 0.80$ ab | $92.54 \pm 17.87$ ab |
| Cristal × Control | $199.52 \pm 38.53$ bc | $7.96 \pm 1.54$ b | $2.55 \pm 0.49$ cd | $90.53 \pm 17.48$ ab |
| Siriana × Nutrifine | $237.76 \pm 45.92$ abc | $8.79 \pm 1.70$ ab | $2.37 \pm 0.46$ cd | $97.88 \pm 18.90$ ab |
| Siriana × Orgevit | $302.47 \pm 58.42$ a | $10.02 \pm 1.93$ ab | $3.51 \pm 0.68$ bc | $117.24 \pm 22.64$ a |
| Siriana × Micoseed | $314.23 \pm 60.69$ a | $12.36 \pm 2.39$ a | $4.82 \pm 0.93$ a | $122.57 \pm 23.67$ a |
| Siriana × Control | $243.64 \pm 47.05$ ab | $9.44 \pm 1.82$ ab | $3.02 \pm 0.58$ bcd | $100.54 \pm 19.41$ ab |
| Signification | * | * | * | * |

Within each column: *—significant differences; values associated with the same lower-case letters are not statistically different at $p \leq 0.05$ according to Duncan's test.

For tomato fruits grown under chemical, organic and biological fertilization 18 polyphenolic compounds (caffeic acid, chlorogenic acid, *p*-coumaric acid, ferulic acid, gentisic acid, sinapic acid, caftaric acid, kaempferol, apigenin, rutin, quercetin, quercetin3-β-D-glucoside, isoquercitrin, fisetin, hyperoside, myricetin, patuletin and luteolin) were quantitative and qualitative analyzed. The analyses performed showed that only 5 polyphenolic com-

pounds: chlorogenic acid, *p*-coumaric acid, ferulic acid, rutin and quercetin were detected in tomato fruits.

The effect of the tomato cultivar and the fertilization type on total polyphenol contents is shown in Table 5. Data present for cultivar did not induce significant differences for *p*-coumaric acid, rutin, ferulic acid, and quercitrin, with the chlorogenic acid exception where the values in the case of the Siriana variety were 211% higher than 'Cristal'. High values among polyphenols were achieved in the case of rutin which is an anticancer compound. The data in the table show that fertilization is a determining factor in the assimilation of polyphenolic compounds with an antioxidant role. High values, in the case of all compounds, were obtained under fertilization with microorganisms both compared to control and compared to conventional chemical fertilization.

**Table 5.** Influence of tomato cultivar and fertilization type on polyphenol compounds.

| Treatment | Chlorogenic Acid ($\mu g \cdot 100$ mL$^{-1}$) | *p*-Coumaric Acid ($\mu g \cdot 100$ mL$^{-1}$) | Rutin ($\mu g \cdot 100$ mL$^{-1}$) | Ferulic Acid ($\mu g \cdot 100$ mL$^{-1}$) | Quercitin ($\mu g \cdot 100$ mL$^{-1}$) |
|---|---|---|---|---|---|
| | Cultivar | | | | |
| Cristal | $17.53 \pm 6.93$ | $1.18 \pm 1.38$ | $82.60 \pm 35.19$ | $4.25 \pm 5.04$ | $0.68 \pm 0.73$ |
| Siriana | $37.12 \pm 13.41$ | $1.94 \pm 1.73$ | $63.68 \pm 24.29$ | $5.58 \pm 5.22$ | $0.87 \pm 0.97$ |
| Signification | * | ns | ns | ns | ns |
| | Fertilization type | | | | |
| Nutrifine | $28.25 \pm 10.27$ a | $0.00 \pm 0.00$ c | $70.00 \pm 13.97$ b | $0.00 \pm 0.00$ c | $0.00 \pm 0.00$ c |
| Orgevit | $32.95 \pm 14.93$ a | $1.95 \pm 0.51$ b | $78.85 \pm 14.12$ b | $5.7 \pm 0.99$ b | $1.30 \pm 0.25$ b |
| Micoseed | $34.95 \pm 15.37$ a | $3.72 \pm 0.94$ a | $107.1 \pm 30.53$ a | $12.3 \pm 2.35$ a | $1.80 \pm 0.45$ a |
| Control | $13.15 \pm 6.87$ b | $0.56 \pm 0.62$ c | $36.60 \pm 9.42$ c | $1.65 \pm 1.85$ c | $0.00 \pm 0.00$ c |
| Signification | * | * | * | * | * |

Within each column: ns—no statistically significant difference; *—significant differences; values associated with the same lowercase letters are not statistically different at *p* < 0.05 according to Duncan's test.

Chlorogenic acid was identified in tomato fruits. The highest values were recorded in 'Siriana' tomato, especially in the biologically and organically fertilized, and similarly for *p*-coumaric acid, which was quantified under the influence of the same types of fertilization, and in the same cultivar (Table 6). The rutin content is present in tomato fruits where the highest values were analyzed in 'Siriana' cv., and fertilized with Micoseed 129.00 $\mu g \cdot 100$ mL$^{-1}$, respectively. The ferulic acid was detected in both varieties with the highest values being determinate in microbiological fertilized.

Quercitin showed a widely changeable content and it was found in traces in half of the tomato treatments.

**Table 6.** Polyphenol compounds detected in tomato fruits.

| Treatment | Chlorogenic Acid ($\mu g \cdot 100$ mL$^{-1}$) | *p*-Coumaric Acid ($\mu g \cdot 100$ mL$^{-1}$) | Rutin ($\mu g \cdot 100$ mL$^{-1}$) | Ferulic Acid ($\mu g \cdot 100$ mL$^{-1}$) | Quercitin ($\mu g \cdot 100$ mL$^{-1}$) |
|---|---|---|---|---|---|
| Cristal × Nutrifine | $20.10 \pm 2.24$ c | tr | $76.30 \pm 8.51$ b | tr | tr |
| Cristal × Orgevit | $20.50 \pm 2.29$ c | $1.60 \pm 0.18$ cd | $82.20 \pm 9.16$ b | $5.60 \pm 0.62$ b | $1.20 \pm 0.13$ b |
| Cristal × Micoseed | $22.20 \pm 2.47$ c | $3.10 \pm 0.35$ b | $85.20 \pm 9.50$ b | $11.40 \pm 1.27$ a | $1.50 \pm 0.17$ b |
| Cristal × Control | $7.30 \pm 0.81$ d | tr | $42.90 \pm 4.78$ cd | tr | tr |
| Siriana × Nutrifine | $36.40 \pm 4.06$ b | tr | $63.70 \pm 7.10$ bc | tr | tr |
| Siriana × Orgevit | $45.40 \pm 5.06$ ab | $2.29 \pm 0.26$ c | $75.50 \pm 8.42$ b | $5.80 \pm 0.65$ b | $1.40 \pm 0.16$ b |
| Siriana × Micoseed | $47.70 \pm 5.32$ a | $4.34 \pm 0.48$ a | $129.00 \pm 14.38$ a | $13.20 \pm 1.47$ a | $2.10 \pm 0.23$ a |
| Siriana × Control | $19.00 \pm 2.12$ c | $1.11 \pm 0.12$ d | $30.30 \pm 3.38$ d | $3.30 \pm 0.37$ c | tr |
| Signification | * | * | * | * | * |

Within each column: *—significant differences; values associated with the same lower-case letters are not statistically different at *p* ≤ 0.05 according to Duncan's test; tr—trace below detection limit 0.5 $\mu g \cdot$ mL$^{-1}$.

### 3.3. Mineral Content

The influence of the tomato cultivar and the fertilization type on the macroelement contents is shown in Table 7. Data present for the cultivar did not induce significant differences for K, Ca, P, and Mg. The dynamics of the content of macroelements show that K > Mg > P > Ca.

**Table 7.** Influence of tomato cultivar and fertilization type on macroelement contents.

| Treatment | K (mg·100 g$^{-1}$ f.w.) | Ca (mg·100 g$^{-1}$ f.w.) | P (mg·100 g$^{-1}$ f.w.) | Mg (mg·100 g$^{-1}$ f.w.) |
|---|---|---|---|---|
| | | Cultivar | | |
| Cristal | 198.67 ± 38.30 | 9.32 ± 2.90 | 10.25 ± 2.94 | 11.10 ± 2.30 |
| Siriana | 203.84 ± 44.37 | 10.93 ± 2.48 | 11.55 ± 2.82 | 11.87 ± 2.60 |
| Signification | ns | ns | ns | ns |
| | | Fertilization type | | |
| Nutrifine | 233.24 ± 41.52 a | 12.20 ± 2.38 a | 13.30 ± 2.55 a | 12.95 ± 2.34 a |
| Orgevit | 206.08 ± 35.84 ab | 11.00 ± 1.90 a | 12.02 ± 2.08 ab | 12.45 ± 2.27 a |
| Micoseed | 197.20 ± 34.07 ab | 10.28 ± 1.86 a | 10.56 ± 1.84 b | 11.35 ± 1.96 ab |
| Control | 168.49 ± 29.27 b | 7.02 ± 2.17 b | 7.07 ± 1.88 c | 9.18 ± 1.62 b |
| Signification | * | * | * | * |

Within each column: ns—no statistically significant difference; *—significant differences; values associated with the same lowercase letters are not statistically different at $p < 0.05$ according to Duncan's test.

The data in the table show that fertilization is a determining factor in macroelement accumulation. The chemical fertilization determines obtaining the highest values for the analyzed macroelements by 25–88% compared to the unfertilized control.

As shown in Table 8, the use of Nutrifine, Orgevit or Micoseed led to an increase in K, Ca, P and Mg content in tomato fruits. The highest content of macrominerals was found in both tomato cultivars under the effect of Nutrifine, followed by Orgevit fertilization.

**Table 8.** Total content of macroelements in tomato fruits.

| Treatment | K (mg·100 g$^{-1}$ f.w.) | Ca (mg·100 g$^{-1}$ f.w.) | P (mg·100 g$^{-1}$ f.w.) | Mg (mg·100 g$^{-1}$ f.w.) |
|---|---|---|---|---|
| Cristal × Nutrifine | 224.20 ± 43.30 | 11.20 ± 2.16 ab | 12.30 ± 2.37 ab | 12.32 ± 2.38 ab |
| Cristal × Orgevit | 202.36 ± 39.08 | 10.90 ± 2.10 ab | 11.84 ± 2.29 ab | 11.80 ± 2.28 ab |
| Cristal × Micoseed | 196.78 ± 38.00 | 9.80 ± 1.90 ab | 10.34 ± 2.00 abc | 11.37 ± 2.20 ab |
| Cristal × Control | 171.32 ± 33.09 | 5.40 ± 1.04 c | 6.50 ± 1.26 c | 8.90 ± 1.72 b |
| Siriana × Nutrifine | 242.28 ± 46.79 | 13.20 ± 2.55 a | 14.30 ± 2.76 a | 13.58 ± 2.62 a |
| Siriana × Orgevit | 209.80 ± 40.52 | 11.10 ± 2.14 ab | 12.20 ± 2.35 ab | 13.10 ± 2.53 ab |
| Siriana × Micoseed | 197.62 ± 38.17 | 10.76 ± 2.08 ab | 10.78 ± 2.08 ab | 11.32 ± 2.19 ab |
| Siriana × Control | 165.66 ± 31.99 | 8.65 ± 1.67 bc | 8.90 ± 1.72 bc | 9.46 ± 1.83 ab |
| Signification | ns | * | * | * |

Within each column: ns—no statistically significant difference; *—significant differences; values associated with the same lower-case letters are not statistically different at $p \leq 0.05$ according to Duncan's test.

Statistically, the Duncan test showed that chemical fertilizer induced positive significant effects on Ca and P contents in tomato fruits compared to the control.

The influence of the tomato cultivar and the fertilization type on microelements contents is shown in Table 9. Data present for the cultivar did not induce significant differences for Cu, Fe, Mn, and Zn. The dynamics of the content of macroelements show that Fe > Zn > Mn > Cu.

**Table 9.** Influence of tomato cultivar and fertilization type on microelement contents.

| Treatment | Cu (mg kg$^{-1}$ f.w.) | Fe (mg·kg$^{-1}$ f.w.) | Mn (mg kg$^{-1}$ f.w.) | Zn (mg kg$^{-1}$ f.w.) |
|---|---|---|---|---|
| | Cultivar | | | |
| Cristal | 0.45 ± 0.10 | 4.53 ± 1.46 | 0.47 ± 0.16 | 4.57 ± 1.82 |
| Siriana | 0.54 ± 0.14 | 5.17 ± 1.54 | 0.56 ± 0.20 | 4.72 ± 1.98 |
| Signification | ns | ns | ns | ns |
| | Fertilization type | | | |
| Nutrifine | 0.52 ± 0.11 ab | 6.36 ± 1.21 a | 0.61 ± 0.13 a | 5.29 ± 0.93 b |
| Orgevit | 0.61 ± 0.13 a | 5.32 ± 0.92 ab | 0.69 ± 0.13 a | 6.51 ± 1.16 a |
| Micoseed | 0.47 ± 0.08 b | 4.65 ± 0.94 b | 0.48 ± 0.10 b | 4.77 ± 0.83 b |
| Control | 0.40 ± 0.07 b | 3.07 ± 0.64 c | 0.29 ± 0.05 c | 2.02 ± 0.35 c |
| Signification | * | * | * | * |

Within each column: ns—no statistically significant difference; *—significant differences; values associated with the same lowercase letters are not statistically different at $p < 0.05$ according to Duncan's test.

The data in the table show that fertilization is a determining factor in microelement accumulation. The organic fertilization determines obtaining the highest values for the analyzed microelements (Cu, Mn, Zn) by 25–88% compared to the unfertilized control with Fe, where the higher content was obtained under Nutrifine fertilization.

Similarly, to the macroelements, the results concerning microelement content revealed that the contents of Cu, Fe, Mn, and Zn in the tomatoes were also affected by the interaction between cultivar and fertilization type (Figure 2).

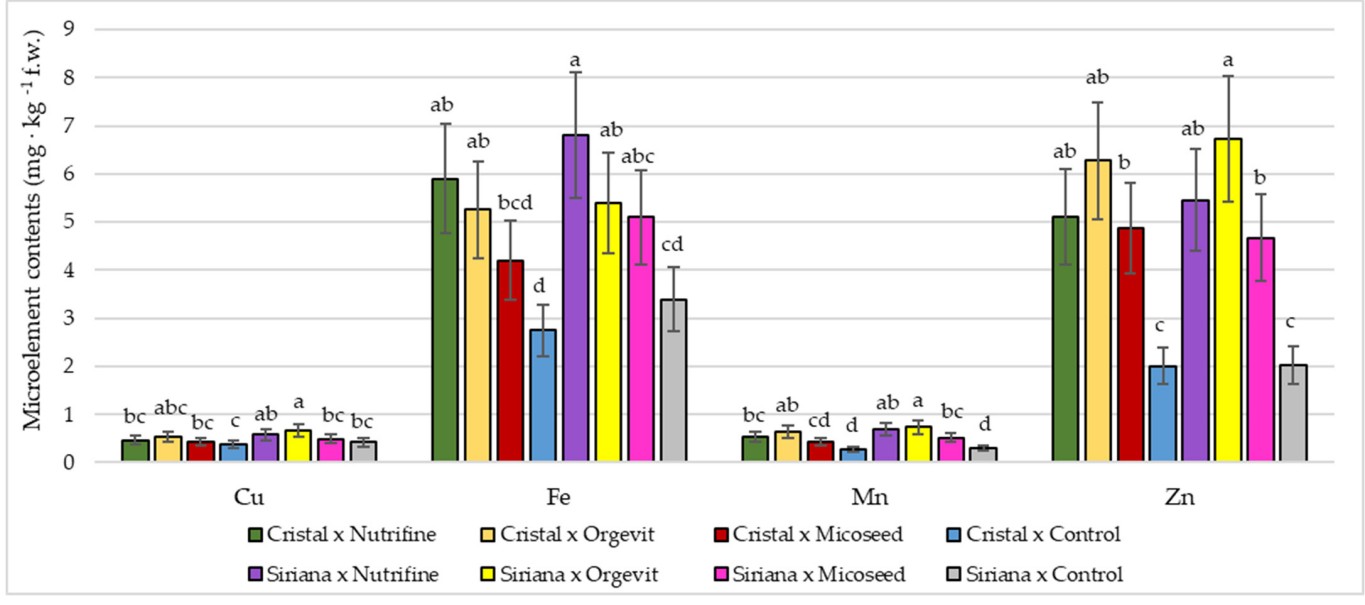

**Figure 2.** Total content of microelements in tomato fruits. Values associated with the same lower-case letters are not statistically different at $p \leq 0.05$ according to Duncan's test.

The use of chemical fertilizer resulted in the highest Fe content in tomato fruits, while the highest content of Cu, Mn and Zn was found when organic fertilizer was applied. Analysis of the microelements content in the studied cultivars showed that the highest quantities of microelements were determined in the fruits of the 'Siriana' cv.

Insignificant differences were observed in 'Cristal' cultivar fruits for Cu content, regardless of applied fertilizer, and for Fe and Mn contents when organic fertilizer was used. The Micoseed use in the fertilization of tomatoes did not cause significant changes in the Cu and Fe contents in the fruits of 'Siriana'.

### 3.4. Biometrical and Yield Parameters

The effect of individual factors on the production characteristics is presented in Table 10. Data presented for both technological factors did not induce significant differences concerning the mentioned parameters (Table 10).

**Table 10.** Influence of tomato cultivar and fertilization type on yield characteristics.

| Treatment | No of Fruits per Plant | Mean Weight per Fruit (g) | Yield (t·ha$^{-1}$) |
|---|---|---|---|
| | Cultivar | | |
| Cristal | 24.1 ± 3.05 | 186 ± 21 | 129.65 ± 16.33 |
| Siriana | 23.9 ± 3.02 | 162 ± 20 | 102.29 ± 12.89 |
| Signification | ns | ns | ns |
| | Fertilization type | | |
| Nutrifine | 24.5 ± 3.07 | 168 ± 21 | 115.07 ± 14.49 |
| Orgevit | 23.6 ± 2.96 | 178 ± 22 | 116.04 ± 14.62 |
| Micoseed | 25.1 ± 3.16 | 183 ± 23 | 129.74 ± 16.34 |
| Control | 23.0 ± 2.90 | 164 ± 20 | 103.01 ± 12.97 |
| Signification | ns | ns | ns |

Within each column: ns—no statistically significant difference for $p < 0.05$ according to Duncan's test.

The data presented for the cultivar, highlight the superiority in terms of harvest for the 'Cristal' variety for the number of fruits per plant, the average weight of the fruits, and the total production. The influence of microbiological fertilization showed that there are differences between the types of fertilization in the case of the three monitored parameters, but the differences were not significant.

The effect of interaction between varieties and nutrition on the production characteristics is presented in Table 11. Data presented for both factors did not induce significant differences for the number of fruits per plant and the average fruit weight. The yield determined by cultivar and fertilization regime increased from 89.33 t·ha$^{-1}$ on 'Siriana' untreated till to 143.42 t·ha$^{-1}$ in version 'Cristal' treated with Micoseed with significant differences for $p \leq 0.05$.

**Table 11.** Influence of interaction of tomato cultivar and fertilization type on yield parameters.

| Treatment | No of Fruits per Plant | Mean Weight per Fruit (g) | Yield (t·ha$^{-1}$) |
|---|---|---|---|
| Cristal × Nutrifine | 24.6 ± 3.11 | 181 ± 23 | 128.75 ± 16.22 ab |
| Cristal × Orgevit | 23.7 ± 2.99 | 191 ± 24 | 129.72 ± 16.34 ab |
| Cristal × Micoseed | 25.2 ± 3.17 | 195 ± 25 | 143.42 ± 18.07 a |
| Cristal × Control | 23.1 ± 2.90 | 175 ± 22 | 116.69 ± 14.70 ab |
| Siriana × Nutrifine | 24.4 ± 3.07 | 156 ± 20 | 101.40 ± 12.78 ab |
| Siriana × Orgevit | 23.5 ± 2.96 | 166 ± 21 | 102.36 ± 12.89 ab |
| Siriana × Micoseed | 25.1 ± 3.17 | 171 ± 22 | 116.06 ± 14.62 ab |
| Siriana × Control | 22.9 ± 2.90 | 154 ± 19 | 89.33 ± 11.25 b |
| Signification | ns | ns | * |

Within each column: ns—no statistically significant difference; *—significant differences; values associated with the same lower-case letters are not statistically different at $p \leq 0.05$ according to Duncan's test.

## 4. Discussion

This research aimed to evaluate the interactions between two cultivars and three fertilization types on the qualitative compounds of tomato fruits, considering the increased interest of consumers to eat healthy food rich in bioactive compounds in the last two decades [42]. Secondly, current farming systems demand more restrictions regarding the use of synthetic chemical origin inputs.

Indeed, secondary metabolites have an essential role in human nutrition as they are used in the physiological processes of growth and in the maintenance of health. These compounds are taken from the food and the deficiency of certain elements leads to disease appearance. Tomato fruits are excellent sources of minerals and antioxidant components for humans. The composition and content of polyphenols, vitamins and minerals in vegetables vary according to the species, variety, phenological stage, and crop management. In this study, the tomato local cultivar that was examined showed a higher fruit nutrient content under the fertilization regime compared to the control.

### 4.1. Water Content, Dry Matter, Total Soluble Solid, Acidity and Ash of Tomato Fruits

As expected, the fruit-dry matter content varied within similar limits reported in previous studies. In addition, the fruit water content was inversely related to the dry matter, being a normal result, taking into account that both parameters constitute the fresh biomass. Regarding the TSS content, apart from reflecting the dry matter of fruits, it is known that together with the acidity, phenols and minerals determine the fruits organoleptic and nutritional quality [43]. In our study, the content of the TSS was not significantly influenced by the fertilization type and the values registered were in the range of those previously reported [44–46]. Along with the TSS, acidity contributes to the flavor and aroma of the tomatoes, and at the same time represents an indicator of the ripeness of the fruits. The lower the citric acid content, the more advanced the level of the fruit ripening process [45,47]. The results of our study showed that for 'Siriana' tomato cultivar treated with biological fertilizer, the amount of citric acid was the highest, indicating an accumulation of this organic acid. The higher content of citric acid in the fruits belonging to the biologically fertilized plants might be due to the presence of microorganisms that either enhanced the uptake of nitrate or modulated nitrate metabolism, and the synthesis of organic acids. Usually, the amount of organic acids in fruits is related to nitrate metabolism. More specifically, during the assimilation of nitrate, the synthesis of carbohydrates decreases and more organic acids are produced [48,49]. In our study, it is likely that the microorganisms present in the fertilizers enhanced the citric acid production and its accumulation in the fruits of tomato. Our results are in agreement with those reported by Inculet et al. [10] via a study conducted on tomato that was inoculated with arbuscular mycorrhizal fungi and bacteria on tomato.

### 4.2. Antioxidant Contents

The content of total polyphenolic compounds, lycopene and β-carotene are improved by using varieties and fertilizers. The new fertilization products obtained on the basis of microorganisms, and especially those from the genus: *Glomus* sp., *Rhizophagus* sp. and *Trichoderma* sp., are superior to chemical and organic fertilization. In tomatoes, regardless of cultivar, 'Cristal' obtained higher values by up to 48% compared to chemical fertilization, 'Siriana' by up to 32% compared to the same fertilization.

Ayuso-Yuste et al. [50] showed that the traditional tomato varieties that were tested, proved richer in lycopene and β-carotene than commercial ones in the last ripening stages.

Increased contents of polyphenols under microbiological fertilization were detected in other species like quinoa [51].

'Siriana' variety had a 39% increase compared to 'Cristal' under the biological fertilization, which suggests both the better adaptation of this local hybrid to the growing conditions and a more effective response to the fertilization input. Similar results were obtained by Mihalache et al. [21] and Spagna et al. [52].

Antioxidant activity levels determined in tomatoes are similar to those reported in the literature. The antioxidant activity values in tomatoes found by Spagna et al. [52] were between 72.84 and 83.05 mmol Trolox·100 $g^{-1}$ d.w.

Consistent with our results from previous research carried out on tomatoes, a beneficial effect of *Bacillus licheniformis* application was recorded on fruit flavonoids and polyphenols content, as well as on antioxidant activity [53].

The quantity and quality of polyphenols from tomato is a positive aspect of a balanced diet, due to the nutrients. Many current studies associate the consumption of tomato with a positive effect on health. Raiola et al. [54] analyzed polyphenols from tomato, and highlighted that fruits lose their nutritive qualities through processing, but the values analyzed for rutin and quercitrin in tomato were lower than our study. The polyphenolic compounds determined in tomato are superior to those obtained by Rosa-Martinez et al. [17] for chlorogenic acid, rutin, and ferulic acid in both species. Through processing, part of the compounds are destroyed, or as the fruits reach over-ripeness, part of these compounds degrade [55,56].

*4.3. Mineral Content*

In the 'Cristal' variety of tomato, the content of K, Ca, P, Mg, Cu, Fe, Mn, and Zn reached a maximum of 224, 11, 12, 12, 0.5, 3, 0.6, 6 mg/100 g f.w., and in 'Siriana' 242, 13, 14, 13, 0.6, 3, 0.7, 6 mg·100 g f.w. The results registered in our study are in agreement with those obtained by Cvijanović et al. [57] who showed that the nutrients content (Ca, K, Mg, P) in tomato were higher in the integrated growing system (1410, 22,300, 1640, and 4300 mg·kg d.w., respectively) compared with those in the organic system (1210, 21,700, 1630, and 3930 mg/kg d.w., respectively). The differences between the two systems are due to the fact that in the integrated system, the use of K and Ca salt based fertilizers are allowed [57]. In our study, the genotype did not significantly influence the content of micro and macro-nutrient. Ciudad-Mulero et al. [58] reported that in tomatoes, the mineral content has varied significantly between the different forms of Yellow tomato, Round Tomato, Long Tomato, and Oxheart Tomato (Na 0.58–3 mg·kg$^{-1}$, Ca 4.4–6.8 mg·kg$^{-1}$, K 1.58–2.15 mg·kg$^{-1}$, 4.5–5.6 mg·kg$^{-1}$, Fe 0.19–0.49 mg·kg$^{-1}$, Cu 0.085–0.14 mg·kg$^{-1}$, Mn 0.023–0.047 mg·kg$^{-1}$, and Zn 0.08–0.345 mg 100 g$^{-1}$ f.w.).

In many studies, it has been shown that the content of secondary metabolites (e.g., polyphenol compounds) in plants is highly dependent on the availability of N to plants. It has been proven that the plant growing in a nitrogen-poor condition leads to the increase of secondary metabolites content in plant parts compared to the level found in plants growing in a nitrogen-rich soil [59,60]. According to Ibrahim et al. [59] the low nitrogen availability in the soil limits the plant growth more than the photosynthesis, and thus, the plants allocate the extra carbon to the production of carbon-based secondary metabolites and not for protein required in plant growth. It is known that the main actions of microorganisms used as biofertilizers are the phosphate and potassium solubilization and nitrogen fixation from soil [61]. Through the use of organic and chemical fertilizers, phosphorus-, potassium- and nitrogen-based compounds are introduced into the soil, sometimes in quantities exceeding the plant needs. Therefore, when using biofertilizers, the plants are grown in a nitrogen-poor condition and the content of secondary metabolites, including in plant fruits, increases. Therefore, these considerations might explain why the content of polyphenol compounds detected in tomato fruits grown under biological fertilization are higher compared to the contents determined in the fruits of plants grown under chemical or organic fertilization.

The higher Fe content in chemically fertilized tomatoes, may be due to the fact that the chemical fertilizer used contains microelements, but also chelating agents (EDTA) that increase the bioavailability of iron to plants [62]. The microorganisms from biofertilizers enhance the Fe uptake and transport it to the plants by secretion of siderophores, whose secretion rate may be affected by environmental conditions such as pH, temperature, nutrient sources, aerobic/anaerobic, etc., [63].

The results of the present research related to local and recently introduced varieties, highlight that the local varieties can be promoted in sustainable crops, due to their ecological plasticity and adaptation to environmental conditions. The presence of some polyphenol compounds, even in small concentrations or traces in the fruits of tomato (ferulic acid, isoquercitrin, gentisic acid), is the premise of a needed continuation of this investigation, combining the different types of fertilization examined, i.e., chemical, organic and biological,

and monitoring the content of macro- and microelements, nitrates and heavy metals to prevent that their contents exceed the allowed thresholds.

In our experiment, the increase of nutritional compounds in tomato fruits, influenced by cultivar and fertilization type, represents a positive impact on consumers' demand for healthy vegetable products.

### 4.4. Biometrical and Yield Parameters

Usually, yield characteristics are determined by the genetic value and the uniformity of the F1 hybrids. 'Cristal' is a tomato hybrid, recently introduced into the crop primarily for its productive qualities and for the fact that it adapts well to the conditions in protected areas, compared to the 'Siriana' hybrid which excels from a qualitative point of view. Similar results for 'Siriana' were also obtained by Inculet et al. [10], which demonstrates that from a quantitative point of view, the hybrid has well-defined biological limits, but with high-ecological plasticity, which makes it recognizable for organic crops.

In general, the nutrition of tomato plants has a determining role in increasing production, it being known that each vegetable species has a specific consumption of nutritional elements. Although the differences between the types of treatments are not significant from a statistical point of view, the production was increased in the biologically fertilized version by up to 25.95% compared to the untreated control.

In general, the number of fruits per plant is a genetic constant, which indicates that the influence of the interaction is less, being strictly biologically determined. The average fruit weight was positively influenced by the cultivar x nutrition interaction from 154 g per fruit in the case of the untreated 'Siriana' cultivar to 195 g per fruit in the case of the biologically fertilized 'Cristal' cultivar; the difference being 26.6%. Both cultivars obtained higher yields in the biologically fertilized variants, up to 22.9% in the case of the 'Cristal' hybrid and up to 29.9% in the case of the local 'Siriana' hybrid compared to the control. The production results obtained in the present study regarding the use of biological fertilization, are in the same way with the results obtained by Bona et al. [49].

In the case of both varieties, no significant production differences were obtained between chemical and organic fertilization, regardless of the combination, which indicates that both cultivations can be used both in conventional and organic systems. Similar results were reported by Terada et al. [64] who examined the effects of soluble chemical and organic fertilizers on the quality and growth of a Micro-Tom tomato. They have found that, for example, the number of leaves, yield, and mean weight per fruit did not differ significantly when the two types of fertilization were applied. Wu et al. [65] investigated the individual and combined effects of chemical and organic fertilizers on growth and fruit yield of tomato cv. 'Changfeng 5', and determined that the total yield was slightly higher when the organic fertilizer was applied. However, statistically, the total yield was not significantly different from that resulted when the chemical fertilizer was applied. Significantly higher yield was obtained when the combination of organic and chemical fertilizer was used.

### 5. Conclusions

Varieties and nutrition measures have a strong influence on the quality and nutritional value of tomatoes. We found that the local tomato hybrid 'Siriana' had a higher rutin content. Moreover, biological fertilization was found to have a positive effect on the accumulation of polyphenols, lycopene, and β-carotene in tomato fruits, which suggests that it can be a viable alternative to chemical fertilization in terms of producing premium quality products. Additionally, organic fertilization increased the nutritional value of tomato, which provides an opportunity for local producers to use organic fertilizers on a large scale. Chemical fertilization was found to increase the total amount of mineral elements in tomato fruits due to its higher solubility, but it also raises concerns about the accumulation of essential heavy metals beyond the allowed thresholds, such as Cu, Zn, Fe, and Mn. Therefore, the content of minerals should be carefully monitored to ensure that they do not exceed the safety limits. Finally, our study highlights the need for further

research on the efficiency of different doses of organic and biological fertilizers to determine their impact on improving the nutritional value of tomato fruits. This will help to optimize the use of these fertilization types and ensure that local producers can produce high-quality and safe products that meet the growing demand for healthier food options.

From the point of view of production, the proposed factors under this study provide information about the nutrition and biological varieties, which can be further used as inputs for sustainable production, especially for organic farming.

**Author Contributions:** Conceptualization, O.-R.R. and V.S.; methodology, G.-C.T., G.C. and A.S., software, A.C. and M.B.; validation, O.-R.R., G.C. and V.S.; formal analysis, I.M., D.A.-M., G.M. and M.R.; investigation, G.-C.T., A.C. and M.B.; resources, V.S. and G.C.; data curation, G.-C.T., M.B. and G.M.; writing—original draft preparation, O.-R.R., I.M., D.A.-M., G.M. and V.S. writing—review and editing, M.R., G.C. and V.S.; visualization, O.-R.R., G.C., A.S. and V.S.; bibliographic, A.C. and O.-R.R.; supervision, V.S., A.S. and G.C.; project administration, O.-R.R. and V.S. All authors have read and agreed to the published version of the manuscript.

**Funding:** This research received no external funding.

**Data Availability Statement:** Not applicable.

**Acknowledgments:** The authors wish to thank "Ion Ionescu de la Brad" Iasi University of Life Sciences for the financial support of the experiments.

**Conflicts of Interest:** The authors declare no conflict of interest.

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
