# Peer review of "Interaction Effects of Cultivars and Nutrition on Quality and Yield of Tomato"

_horticulturae, doi:10.3390/horticulturae9050541_

Round 1

Reviewer 1 Report (New Reviewer)

Varieties and fertilization management have very important effects on the nutritional value of tomato. This study aimed to improve the nutritional value of two tomato varieties with three fertilization types. The results showed that primary metabolism compounds do not vary significantly with different type of fertilizer used. However, the results for the antioxidant compounds showed a better effect of biological fertilization compared to chemical fertilizer and control unfertilized. The tomato fruits from the local cultivar (‘Siriana’) are richer in nutritional compounds regardless of the type of fertilization, which denotes a good ability to adapt to crop conditions. Furthermore, tomato cultivars reacted positively to microbiological fertilization compare to chemical, thus producing nutritious fruits under sustainable management.

In general, this manuscript is well written and experimental design is good and produced  significant results. I have only the following minor comments:

1.       For the chemical fertilizer treatment, The NPK chemical fertilization was performed with 200 kg·ha-1 of Nutrifine® 153 NPK 20-20-20, applied before transplant, and two supplies of 300 kg·ha-1 of Nutrifine ® 154 NPK 9-18-27+2 MgO. Any explanation for it?

2.       Table 1 and Table 3, the digital numbers are not the same, should all keep two digital number, if it is 0, it also should put it on.

3.       For the figure 2. Why author put the data on the bar, only a b and c show the significant would be good enough.

4.       Literature should be updated, there is no literature from 2023.

Author Response

Dear Reviewer 1

We thank you again for your interest in our paper. Our point-by-point responses regarding comments are detailed on the following pages. All the changes suggested were highlighted with yellow.

Varieties and fertilization management have very important effects on the nutritional value of tomato. This study aimed to improve the nutritional value of two tomato varieties with three fertilization types. The results showed that primary metabolism compounds do not vary significantly with different type of fertilizer used. However, the results for the antioxidant compounds showed a better effect of biological fertilization compared to chemical fertilizer and control unfertilized. The tomato fruits from the local cultivar (‘Siriana’) are richer in nutritional compounds regardless of the type of fertilization, which denotes a good ability to adapt to crop conditions. Furthermore, tomato cultivars reacted positively to microbiological fertilization compare to chemical, thus producing nutritious fruits under sustainable management.

In general, this manuscript is well written and experimental design is good and produced significant results. I have only the following minor comments:

Comment 1.  For the chemical fertilizer treatment, The NPK chemical fertilization was performed with 200 kg·ha-1 of Nutrifine® NPK 20-20-20, applied before transplant, and two supplies of 300 kg·ha-1 of Nutrifine ® NPK 9-18-27+2 MgO. Any explanation for it?

Answer to Comment 1. The chemical fertilizer formula from the Nutrifine range was determined by the phenophase of growth and development in correlation with soil analyzes and specific consumption. At the beginning, when preparing the soil, a balanced recipe was used for NPK and then with higher requirements for P and K than nitrogen, as it is necessary for the formation, development and ripening of the fruits. The first application was made three weeks after planting and the second application three weeks after the first application.

Correction. Line 185 – BBCH 803-805 not 703-705 

Comment 2.  Table 1 and Table 3, the digital numbers are not the same, should all keep two digital number, if it is 0, it also should put it on.

Answer to Comment 1. Add

Comment 3.  For the figure 2. Why author put the data on the bar, only a b and c show the significant would be good enough.

Answer to Comment 1. Add

Comment 4.  Literature should be updated, there is no literature from 2023.

Answer to Comment 1. The references list was updated with references from 2023 (references number 19, 64 and 65) and consequently, a new paragraph was added (see lines 540-550).

Reviewer 2 Report (New Reviewer)

Submitted falls into the scope of a horticulture journal and I found it an interesting and relatively well-written paper. The article is focus on the Interaction effects of cultivars and nutrition on the quality and yield of tomato

 There are some suggestions with the manuscript which need to be addressed

 The abstract is well written, however, in line 26 authors showed “the results for the antioxidant compounds effect of biological fertilization compared to chemical fertilizer” but did not mention percentages of how small or large antioxidants amount increased and if possible other results values also mentioned in the abstract.

 The keywords are appropriate for the manuscript

 The introduction section is written well for the manuscript       

  In the methodology section, the authors showed soil characteristics N and P while K is missing. However, the authors do not mention that it is N, P total nitrogen, or available mineral nitrogen same case for P it is available P content or total P? What is the reason for putting the (.) full stop punctuation between (mg.kg-1)?

Table 3, line, 299 for both cultivars have significant differences which showed in the table but not mentioned in lowercase letters in the table.

Line 314, 115, Table 4 showed significant differences “values associated with the same lower-case letters are not statistically different at p ≤ 0.05 according to Duncan’s test” Please check whether the same lowercase letters or different lowercase letters and results have shown significant differences where authors mention are not statistically different.

Please make corrections throughout the manuscript same, like the table, 4 please check the whole table notes lines 334 and 335, and 346,347…

 NS did not significantly show the same pattern of Ns or ns throughout the manuscript.

 It’s better to mention the Table 4 results in the result section in % value line 305,306 because 152.47 μg•100 mL-1 results value is already shown in table form.

  The discussion and conclusion sections are written well

Author Response

Dear Reviewer 2

We thank you again for your interest in our paper. Our point-by-point responses regarding comments are detailed on the following pages. All the changes suggested were highlighted with green.

Submitted falls into the scope of a horticulture journal and I found it an interesting and relatively well-written paper. The article is focus on the Interaction effects of cultivars and nutrition on the quality and yield of tomato

 There are some suggestions with the manuscript which need to be addressed

 Comment 1.  The abstract is well written, however, in line 26 authors showed “the results for the antioxidant compounds effect of biological fertilization compared to chemical fertilizer” but did not mention percentages of how small or large antioxidants amount increased and if possible other results values also mentioned in the abstract.

Answer to Comment 1. Add. Thus, the antioxidant activity was improved by 28% compared to chemical fertilization, the lycopene content by 36% and β-carotene by 96%, respectively. The keywords are appropriate for the manuscript

 The introduction section is written well for the manuscript       

 Comment 2.  In the methodology section, the authors showed soil characteristics N and P while K is missing. However, the authors do not mention that it is N, P total nitrogen, or available mineral nitrogen same case for P it is available P content or total P? What is the reason for putting the (.) full stop punctuation between (mg.kg-1)?

Answer to Comment 2. Add -  5.91 C/N 2.8 g·kg-1 N, 32 mg·kg-1available P, 218 mg·kg-1 available K

Comment 3.  Table 3, line, 299 for both cultivars   have significant differences which showed in the table but not mentioned in lowercase letters in the table.

Answer to Comment 3. In general, for comparing only two factors it is not necessary to use lowercase letter, because it is understood that there is significance between the graduations of the cultivar factor. If this is the case, we can use the meaning, but in general it is not used.

Comment 4.  Line 314, 115, Table 4 showed significant differences “values associated with the same lower-case letters are not statistically different at p ≤ 0.05 according to Duncan’s test” Please check whether the same lowercase letters or different lowercase letters and results have shown significant differences where authors mention are not statistically different.

Answer to Comment 4. If there is the same letter for two or more values, it means that there are no significant differences between the combinations of factors. This aspect was also accepted by the other 2 reviewers

Comment 5.  Please make corrections throughout the manuscript same, like the table, 4 please check the whole table notes lines 334 and 335, and 346,347…

Answer to Comment 5. Add

 Comment 6.  NS did not significantly show the same pattern of Ns or ns throughout the manuscript.

Answer to Comment 6. Rewrite

 Comment 7.  It’s better to mention the Table 4 results in the result section in % value line 305,306 because 152.47 μg•100 mL-1 results value is already shown in table form.

Answer to Comment 6. Yes. It was improved. The paragraph was rewrite

  The discussion and conclusion sections are written well

Reviewer 3 Report (New Reviewer)

Dear authors,

Attached are the comments. Please be specific and have clarity of the context.

Please make simple and possibly short statements.

In scientific writing, clarity is important.

Avoid any unnecessary paragraphing and repetition of information.

Author Response

Dear Reviewer 3

We thank you again for your interest in our paper. We also thank the reviewers for the patient and careful examination of our manuscript and for providing ideas and corrections that will improve this manuscript. Our point-by-point responses regarding comments are detailed on the following pages. All the changes suggested were made with blue.

Comment 1: Line 50-51: make it as a one paragraph. So, it will maintain the flow of the context

Answer to Comment 1: The change was done as recommended.

Comment 2: Line 58-59: Syntax error. Confusing. Rewrite the statement. May be write it in two statements. For example: Tomato fruits; rich in all macro- and micronutrients bring balanced elemental composition essential for synthesis of primary and secondary metabolites, which also improves organoleptic and biochemical qualities of the fruits. (does this statement change the meaning?)

Answer to Comment 2: The statement was rewrite as suggested.

Comment 3: Line 59-60; What are later compounds? Please clarify

Answer to Comment 3: To be clearer, “later compounds” was replaced with “secondary metabolites”

Comment 4: Line 69-73: Make sure it is one paragraph

Answer to Comment 4: The change was done as recommended

Comment 5: Line 74: Make simple statement structure.

e.g. Fertilizers play the main role in enhancing the nutritional status of plants or Fertilizers

improve nutritional status of plants.

Answer to Comment 5: The change was done as recommended: “Fertilizers play the main role in enhancing the nutritional status of plants”

Comment 6: Line 75: write a simple and short statements unless otherwise is required.

Answer to Comment 6: “Chemical fertilization has proven to be a method that causes numerous harmful effects on the environment and human health.” was rephrased as follows: “Chemical fertilization is a widely used method for increasing crop yields, but it has been found to have significant negative impacts on the environment and human health”

Comment 7: Line 90: that, remove comma. Cite the research. Why are you mentioning diced tomatoes? Is it relevant?

Answer to Comment 7: The comma was removed.

Regarding the dived tomatoes, thank you for your remark. The paragraph was removed (Research carried out on processing tomato highlighted that compared to the raw fruits, the diced product showed an increase in some important quality parameters, such as total and soluble solids, reduced sugars and antioxidants, except for lycopene which was not affected by the industrial treatment [24].)

Comment 8: Line 74-96: need to rewrite to avoid repetitive information and maintain the flow of context. Tomato intake provide sufficient nutrient uptake is mentioned in two different paragraphs. Please be specific.

Answer to Comment 8: To maintain the flow of the context, the parapgraph: “In addition to polyphenol compounds, tomato fruits have a considerable content of vitamins (Vit. A, C, K, vitamin B complex) and minerals (Fe, Mg, Zn, K, Ca, P, Mn, Cu), whose level depends on variety, cultivation system, fertilizer type and harvest time [10,25].” was moved to lines 73-76.

Comment 9: Line 128: use end before comma. Or you can mention that these varieties are commercially used in Romania due to their geographical adaptability, resistance to disease and pests, and higher net return advantages over others.

Answer to Comment 9: The statement was rephrased as follows: “Varieties selection was done based on their adaptability to the environmental conditions of Romania, their resistance to pests and their higher net return advantages.”

Comment 10: Line 131: 100 days of what?

Answer to Comment 10: 100 days of germination

Comment 11: Line 132: undetermined growth- is it a category? You may want to then capitalize first letter

‘Undermined growth’ I think it is Indeterminate and not Undetermined. Please check the proper

classification and writing rules.

Answer to Comment 11: It is Indeterminate Growth. The change was done also in the text.

Comment 12: Line 146: Experimental design, plants and treatments What was elemental composition of unfertilized media? Is it chernozem loam-clay? Please mention about your untreated control treatment media/soil characteristics. This question arises because I believe that unfertilized treatment will have some nutrition in its soil composition. Also, in biological treatment, apart from microbial agents, what was mineral nutrient content of your media/soil. This question is important because microbes do not serve as elemental nutrients. They help to make nutrients available. It is supplement and not a fertilizer unless you mention the mineral nutrients supply from your O treatment. Can you fit all your treatment information and how much mineral nutrients supplied in the table format that may give clarity of your treatments?

Answer to Comment 12: The soil is a loam-clay chernozem. Soil analyzes are described in paragraphs 131-134. The quality of microorganisms is that they provide nutrients from the soil complex. This study also created the opportunity to propose in the future the performance of mixed fertilizers organic x microorganism, chemical x microorganism; organic x chemical x microorganisms. The soil samples are average samples collected from the greenhouse before the establishment of the experiment, cultivated for 15 years only with vegetables, where fertilization was carried out uniformly every year organically. For 2 years, the experiment was established in the same solar and on the same plot, so the plants benefited uniformly from the same mineral elements at the beginning of the experiment. This unilateral decision to fertilize only chemically, organically or biologically was determined by the high content of the soil in organic matter.

Comment 13: Line 181: what is AOAC, 2005. Provide manufacturer information

Answer to Comment 13: It is not the manufacturer, but the abbreviation of Association of Official Agricultural Chemists from where the method was used. Usually it is written like this, being accepted by researchers.

Comment 14: Line 190: are 16 samples were pooed together for analytical quantification? Why is it single statement paragraph? Please rewrite how many biological replicates used. Also, HPLC detect the compounds. Therefore, the correct statement should be: HPLC detected eighteen phenolic compounds from the fruits

Answer to Comment 14: The paragraph from the lines: 188-189 “The content of eighteen phenolic compounds in 16 samples of tomato fruits was determined.” Was deleted.

To be clearer the following amendments were done: lines 199-204 “The individual phenolic compounds were determined using RP-HPLC with a UV detector coupled with MS. The detection and quantification of polyphenols was done in triplicate. A number of 17 standard solutions were used to carry out the analyses: caffeic acid, chlorogenic acid, p-coumaric acid, ferulic acid, gentisic acid, sinapic acid, caftaric acid, kaempferol, apigenin, rutin, quercetin, quercetin3-β-D-glucoside, hyperoside, myricetin, isoquercitrin, fisetinpatuletin, luteolin.”

Comment 15: Line 264 and table 1:

You have already shortened/abbreviated names for each treatment such as biological fertilization is (B), chemical is (Ch). Why are you jumping to product name now? Be consistent.

Answer to Comment 15: We would prefer to use the full name, so we deleted the abbreviations.

Comment 16: Line 269: What is Cu parameter?

In results you mention everything as a treatment. Therefore, you can say that O treatment had higher TSS; 18.03 as compared to the control.

Answer to Comment 16: “cu” was written by mistake; it is a typo mistake. We deleted it.

The changes were done as suggested.

Comment 17: Line 291: what is autohtone? “it can be promoted with good results regarding the activity of antioxidant compounds” This statement should go in discussion and not results.

Answer to Comment 17: “autohtone” was replaced with “native”.  The statement: “it can be promoted with good results regarding the activity of antioxidant compounds” was deleted.

Almost the same statement it is already mentioned in discussion section lines: 462-464

Comment 18: Table 5: You mentioned in M&M that HPLC detected 18 polyphenols. Can you show the HPLC peaks? Why data on only 5 polyphenols is presented?

Answer to Comment 18: The paragraph from M&M stating “The content of eighteen phenolic compounds in 16 samples of tomato fruits was determined.” Was deleted because was not clear.

We used a set of 17 standard solutions, but only 5 were found to be present in our tomatoes.

Comment 19: Figure 5: Stick to the table format. Micro-elements graph bar looks odd when you have tables for interaction.

Answer to Comment 19: In principle, I kept Figure 2 in the form of a graph, with the mention on the bar graph only of the significations, having the confirmed acceptance of the other two reviewers.

Round 2

Reviewer 3 Report (New Reviewer)

Dear Authors,

Thank you for clarifications provided. 

This manuscript is a resubmission of an earlier submission. The following is a list of the peer review reports and author responses from that submission.

Round 1

Reviewer 1 Report

In the attached file

Author Response

Dear Reviewer,

Thank you very much for your valuable recommendations and comments. We have carefully considered all your comments and recommendations and we have made changes in the manuscript.

All modifications were highlighted in red color.

Line 2: “cultivars” – please write in one line

We did the change.

Line 34: this sentence does not fit here – the purpose of the research was not to indicate which vegetable has a better chemical/nutritional composition

This sentence was deleted.

Line 36: I propose to change the order of keywords – from vegetables, preparations used and what they influenced / what was tested

The keywords have been rearranged as you suggested

Line 44: “2017” – do you have more recent data?

Unfortunately, we have not found more recent data on the effects of insufficient fruit and vegetable consumption. Even though it is one of the world's major problems, reports from competent authorities on the subject are quite scarce or not available to everyone.

Line 107: “TEAC” – each abbreviation should be explained

The abbreviations were explained.

Line 133: I lack information on how many repetitions were performed for each experiment/analysis

Add

Line 136: should be “Capsicum annuum L.”

We did the change.

Line 138: this sentence: “The tomato cultivars chosen in this study were ‘Cristal’ and ‘Siriana’, respectively ‘Blancina’ and ‘Brillant’…” is not clear, please rewrite. Pepper is not mentioned in this sentence

The phrase was rewritten.

Line 148: TSWV – each abbreviation should be explained

The abbreviation was explained: tomato spotted wilt virus (TSWV)

Line 152: OM – each abbreviation should be explained

The abbreviation was explained: organic matter (OM)

Line 153: ppm is a colloquial unit; for solid samples it is better to use mg/kg

We did the change as suggested.

Line 179: on what basis were the doses of applied fertilizers – chemical and organic – determined? What was this active substance?

Explained into the paragraph

Line 182: on what basis was the dose of biological fertilizer determined, so that the study groups could then be compared with each other? This should be clearly explained

Explained into the text

Line 189: how were plant diseases controlled? What preparations were used? Were agents (pesticides) used for chemical fertilization? It's not clearly written

This sentence was rewrite

Line 199: In my opinion, this fragment “The Proximate Composition of ash was done as follows” is redundant

This sentence was rewrite

Line 206/211: please write correctly symbol (°C) for the degree Celsius, please correct in the whole manuscript

We did the changes as suggested in the whole manuscript.

Line 209: you should list here phenolic compounds, which were determined by HPLC

Phenolic compounds determined are presented into the paragraph 224-227

Line 222: What does “etalon” mean?

Thank you for your observation. The word “etalon” was replaced with “standard solutions”.

Line 240/251: standardize the spelling of units throughout the work: μg/mL or μg.mL-1

We did the change as suggested in the whole manuscript.

Line 248: dot at the end of the sentence

We add the dot at the end of sentence.

Line 249: lack of dot after “Activity”

We delete the dot as suggested.

Line 250: this part “…at three different concentrations of methanol and ethanol extracts: 0.05 mL of 10, 5, and 2.5 mg·mL−1 extracts…” is not clear for me…For which samples did you determine antioxidant activity?

The antioxidant activity was determined for six samples of tomato cultivars (each cultivar was combined with three fertilization) and six samples of pepper cultivars (each cultivar was combined with three fertilization).

Line 256: I do not see results on “The percentage of free-radical scavenging activity” in this manuscript

Add

Line 258: what does QE mean? I see that later in the results Antioxidant Activity is expressed as mmol Trol·100 g−1 d.w. Do you mean – Trolox? There is no consistency here

Rewrite

Line 262: for solid samples usually “content” is used expressed, for example in mg/kg, for liquid samples – “concentration” expressed for example in mg/L. Preferably add the elements that have been analyzed here. What system was used to mineralize the samples? What were the samples mineralized with (mineral acid)? For which samples 0.5 g is used and for which 1 gram? Why this difference? Please add/explain

The macroelements (K, Ca, P, Mg) and microelements (Cu, Fe, Mn, Zn) were determined for both tomato and pepper fruits. After oven drying step, the samples were extracted in an acid mixture (HCl and HNO3), following the reported procedure by Fernández-Ruiz et al (reference 51).

Line 267: you present the data as a mean – has the distribution been checked? Did you have a normal distribution?

Thank you very much for the suggestion, for this study the distribution was not checked but we will do that for our next study.

Line 283: do the tables show average values from two examined years? I don't see that information

We included that information in the Statistical method section.

Line 311: In Table 3, total phenolic is expressed as (μg·100 mL-1), whereas in the methodology it was written “The total phenolic compounds were determined by the Folin-Ciocâlteu method… The results were expressed as mg gallic acid equivalent (GAE)/100 g fresh weight (FW)” – again lack of consistency

We corrected on the text: The results were expressed as mg gallic acid equivalent (GAE)/100g fresh weight (FW) with The results were expressed as of μg of gallic acid equivalents per mL (μg•100mL-1).

Line 311: why the content of lycopene and β-carotene was determined in dry weight, not in the fresh mass? The methodology for these two compounds is very general (Line 244).

Thank you for your remark. Unfortunately, there was an editing error when the manuscript was written. The lycopene and β-carotene content was determined from fresh biomass. The methods used have been described in other studies and, to avoid plagiarism, we have referred to these studies without describing the methods.

Line 311: According to the methodology, free-radical (DPPH) scavenging activity should be calculated for antioxidant activity, using QE (I suppose quercetin) as a control, not “Trol” as indicated in the Table, again lack of consistency

For the antioxidant activity, Trolox equivalent antioxidant capacity (TEAC) of the extracts was calculated by comparing their percentage of free-radical scavenging activity of DPPH with the Trolox standard curve.

Line 328: “p-coumaric acid” – “p” in italics; change also in Table 5

We did the changes as suggested in the whole manuscript.

Line 332: should be “isoquercetin”

We did the change.

Line 337: dot at the end of this sentence, not semicolon

We add the dot at the end of sentence.

Line 341: “trace” is not clear for me. Were these compounds detected in the analyzed samples or were they below the limit of detection? It may be worth specifying the detection limit for these compounds

We added the detection limit 0.5 µg∙mL−1

Line 356: “…but traces were detected only in peppers…” – “traces” is not clear

We added the detection limit 0.5 µg∙mL−1

Line 386: it is not clear to me why the microelement content is shown in the Figure. In this case, the table is more readable. Additionally, the unit now is mg/kg, which makes it difficult to compare results, also in the section - Results and Discussion

The paper has been previously reviewed by other reviewers and at their suggestion the microelement content has been graphically displayed using the unit of measurement mg/kg.

Line 419: The discussion focused on comparing the results with the literature data - what increased, what decreased. I am missing an explanation why a given type of fertilizer caused an increase/decrease in the content of a given compound

Thank you for your suggestion. In the paragraph between the lines 543-565 we gave some explanations on the increase of polyphenols and Fe content in tomato and pepper fruits in relation to the type of fertilizer used

Line 468: dot at the end of this sentence - please pay more attention to punctuation

Line 562: I don't understand the reference to heavy metals here. They were not analyzed in this work. Did the fertilizers used contain heavy metals?

This phrase mainly concerned the essential heavy metals, such as Cu(copper), Fe (iron), Zn (zinc) and Mn(manganese) which have been analyzed. According to metals classification, the metals with density of more than 5 g/cm3 are heavy metals. Therefore, these metals belong to this category. Thus, to avoid confusion, the phrase was completed: “...with particular reference to essential heavy metals (e.g. Cu, Zn, Fe and Mn).

Line 578: All Latin names should be written in italics – e.g., Line 590, 697, 708, 718, 737, 747…etc. Additionally, the second part of the name should be written in lower case – e.g., Solanum lycopersicum

Thank you for your comment. References have been inserted with Zotero and sometimes such mistakes occur. I made the changes manually and hope that the changes will be keept, as we have not yet unlinked the citations.

I do not understand what the results in the Supplementary material represent – I don't know how to read them. This should be better explained.

The additional material was write at the earlier request of another reviewer in order to justify the individual influence of the two authors studied. Also, the volume of material would have been too large for a manuscript

Reviewer 2 Report

The submitted paper to “Horticulturae” describes the interactions between two cultivars of Tomato (Siriana & Cristal) and two cultivars of pepper (Blancina & Brillant) and three fertilization treatments:(1) chemical fertilization, (2) organic fertilization, and (3) biological fertilization, on the qualitative compounds of tomato and pepper fruits including and total soluble solids, ash, phenolic compounds, lycopene β-Carotene and mineral content in addition to the their acidity  and antioxidant activity.

However authors should clearly demonstrate the originality and the relevance/ interest of their findings

Abstract :

Line 28 : please to write as : “a better effect  of biological treatment or fertilization”.

Line 24: please to correct name “Siriana”.

Line 33: “……reacted positively…..:” in comparison to which treatment?

Line 34 & 35: please to rewrite in a comprehensive way, is this related to vegetable nature, or fertilization?

Introduction

Line 61-63 : please to rewrite.

Line 73: please to correct, something in lacking “..portion of 100 g of ??? 73 provides…”.

Line78  : please to correct, something in lacking: “…….and anti-inflammatory effect, ??? recommend that tomatoes…..”

Line78  : Please to change “offers” by “causes”.

Line 106 & 107  : Please to write the complete name of these compounds followed by their abbreviation in brackets:  (DPPH) & (TEAC) for the first time they appear in the text.

Line 112-114: please to rearrange text withing material of the paragraph : Line 119-127.

Line 123: “Assefa and Tadesse”, write the number of the reference just after the names.

Line 129: please to rise clearly the “problematic” and precise “originality of this work” in light of the above mentioned bibliography before the “aims”.

Materials and Methods

Line 136: delete “-“  in “Capsi- cum”.

Line 148: “TSWV” : see remarks for “DPPH” & “TEAC”.

Line 163: correct as “For both tomato and pepper”.

Line 163-171: please to make one paragraph.

Line 203: What’s the meaning of “BBCH 703-705 ».

Phase to give precision for “methanol: acetic acid”, what do you mean by 0.1%?.

Line 22: You have 17 standards, not 16, please begin the sentence with words.

Line 258: “QE”: see remarks for “DPPH” & “TEAC

Results

Line 227: please to use “TSS” for “total soluble solids” in the rest of the manuscript.

Line 325: correct as “was higher”

Line 327-333: Please to rewrite the paragraph. Please check for “Siriana” Characters

Line 328: in M & M you mentioned only 17, please to correct.

Line 345-346: Please to rewrite the paragraph

Line 453-454: Please to write number of the reference after author’s name.  

Line 474-475 delete the sentence unless there is a relationship with phenolics. “Tundis et al. [63] also reported higher levels of carotenoids in red pepper fruits than in yellow peppers”

Line 476-478 : please to rewrite this paragraph

Line 498: Please to write number of the reference after author’s name.  

Figures:

Please to give standard errors in the figure 1.

Please to add a photo illustrating most relevant tomato and pepper results regarding Biological fertilization in comparison to other treatments.

A PCA figure should be presented as to compare and study correlations between treatments and measured parameters and

Author Response

Dear Reviewer,

Thank you very much for your valuable recommendations and comments. We have carefully considered all your comments and recommendations and we have made changes in the manuscript.

All modifications were highlighted in blue color.

The submitted paper to “Horticulturae” describes the interactions between two cultivars of Tomato (Siriana & Cristal) and two cultivars of pepper (Blancina & Brillant) and three fertilization treatments:(1) chemical fertilization, (2) organic fertilization, and (3) biological fertilization, on the qualitative compounds of tomato and pepper fruits including and total soluble solids, ash, phenolic compounds, lycopene β-Carotene and mineral content in addition to the their acidity  and antioxidant activity.

However authors should clearly demonstrate the originality and the relevance/ interest of their findings

Abstract :

Line 28 : please to write as : “a better effect  of biological treatment or fertilization”.

Your suggestion was inserted in the text.

Line 24: please to correct name “Siriana”.

The name was corrected.

Line 33: “……reacted positively…..:” in comparison to which treatment?

The sentence was rewritten.

Line 34 & 35: please to rewrite in a comprehensive way, is this related to vegetable nature, or fertilization?

This sentence was deleted.

Introduction

Line 61-63 : please to rewrite.

The sentence was rewritten.

Line 73: please to correct, something in lacking “..portion of 100 g of ??? 73 provides…”.

100g of tomato was added in text.

Line78  : Please to change “offers” by “causes”.

We did the change as suggested. We rewritten the paragraph. In many studies it is reported that some bioactive substances found in tomatoes, especially lycopene, have anti-inflammatory effects in the human body.

Line 106 & 107  : Please to write the complete name of these compounds followed by their abbreviation in brackets:  (DPPH) & (TEAC) for the first time they appear in the text.

The abbreviations were explained.

Line 112-114: please to rearrange text withing material of the paragraph : Line 119-127.

The text was rearranged

Line 123: “Assefa and Tadesse”, write the number of the reference just after the names.

We did the changes as suggested.

Materials and Methods

Line 136: delete “-“  in “Capsi- cum”.

We did the changes as suggested.

Line 148: “TSWV” : see remarks for “DPPH” & “TEAC”.

The abbreviations were explained.

Line 163: correct as “For both tomato and pepper”.

We did the changes as suggested.

Line 163-171: please to make one paragraph.

We did the changes as suggested.

Line 203: What’s the meaning of “BBCH 703-705 ».

Add comment

Phase to give precision for “methanol: acetic acid”, what do you mean by 0.1%?.

The elution system contains 0.1% acetic acid. We corrected on the manuscript.

Line 22: You have 17 standards, not 16, please begin the sentence with words.

We corrected the number of standards used. In total were used 18 standards.

Line 258: “QE”: see remarks for “DPPH” & “TEAC

Is mentioned above

Results

Line 227: please to use “TSS” for “total soluble solids” in the rest of the manuscript.

We did the changes as suggested.

Line 325: correct as “was higher”

We did the changes as suggested.

Line 327-333: Please to rewrite the paragraph. Please check for “Siriana” Characters

We rewritten the paragraph and modified the characters for “Siriana”

Line 328: in M & M you mentioned only 17, please to correct.

We did the changes as suggested.

Line 345-346: Please to rewrite the paragraph

We rewritten the paragraph.

Line 453-454: Please to write number of the reference after author’s name.  

We did the changes as suggested.

Line 474-475 delete the sentence unless there is a relationship with phenolics. “Tundis et al. [63] also reported higher levels of carotenoids in red pepper fruits than in yellow peppers”

We deleted this sentence.

Line 476-478 : please to rewrite this paragraph

We rewritten the paragraph

Line 498: Please to write number of the reference after author’s name.  

We did the changes as suggested.

Figures:

Please to give standard errors in the figure 1.

The data in figure 1 represent average monthly values for temperature and humidity and maximum values for light intensity. In such article, the calculation of standard errors is not justified because there is the same oscillation during the vegetation period. This aspect would have been justified in the error calculation if we had had multi-year averages or if predictions had been made for the three factors

Please to add a photo illustrating most relevant tomato and pepper results regarding Biological fertilization in comparison to other treatments.

Thank you for your suggestion, but unfortunately, we do not have photos to illustrate the differences between tomatoes or peppers depending on the type of fertilizer.

A PCA figure should be presented as to compare and study correlations between treatments and measured parameters and.

Thank you for your suggestion, is a pretty good idea for highlighting the relationship between treatments and related parameters, which we will try to use in our future studies.

Round 2

Reviewer 1 Report

Generally, the manuscript was improved according to my suggestions, but I have a critical comment to the statistical analysis.

In this type of research, presented in this publication, statistical analysis of the obtained research results is essential/obligatory. Per my comment - Line 267: you present the data as a mean - has the distribution been checked? Did you have a normal distribution?

I got the answer:

Thank you very much for the suggestion, for this study the distribution was not checked but we will do that for our next study.

Checking the normality of the distribution of variables is the first and basic step in choosing an appropriate statistical test. It decides whether we present the results as the mean or median. This cannot be done in the next study, it should be checked in this study.

Otherwise, the presented results are unreliable.

If there was no normal distribution of variables, the results should be reported as the median. I cannot accept such an answer. Now I have serious doubts about these results, whether they are properly presented.

Author Response

Dear Reviewer,

Thank you very much for your valuable recommendations and comments. We have carefully considered all your comments and recommendations and we have made changes in the manuscript. Al new changes made by green color

Generally, the manuscript was improved according to my suggestions, but I have a critical comment to the statistical analysis.

In this type of research, presented in this publication, statistical analysis of the obtained research results is essential/obligatory. Per my comment -

I got the answer:

Thank you very much for the suggestion, for this study the distribution was not checked but we will do that for our next study.

Checking the normality of the distribution of variables is the first and basic step in choosing an appropriate statistical test. It decides whether we present the results as the mean or median. This cannot be done in the next study, it should be checked in this study.

Otherwise, the presented results are unreliable.

If there was no normal distribution of variables, the results should be reported as the median. I cannot accept such an answer. Now I have serious doubts about these results, whether they are properly presented.

Line 267: you present the data as a mean - has the distribution been checked? Did you have a normal distribution?

Authors: At the time of the statistical analysis, it is true that we did not perform the normal distribution.

Following your review, the normal distribution was performed using the Shapiro-Wilk test in SPSS, which is why we thank you.

If necessary, the values for this test can be attached. The data obtained for the Shapiro-Wilk test are normally distributed.

Reviewer 2 Report

In the end of the Introduction section, authors should clearly discuss problematics, present the state of the art. The nature of the fertilizers used in this work should be given at this stage  and clearly explain what is the novelty of their work, originality compared to previous works.

Conclusion: Authors should avoid a "summary" and clearly explain what their work relevance for the progress in the field.

It's regrettable to know that with such a work, performed with two plants, no  pertaining photos that are self explaining are available.

Author Response

Dear Reviewer,

Thank you very much for your valuable recommendations and comments. We have carefully considered all your comments and recommendations and we have made changes in the manuscript.

All modifications were highlighted in green color

In the end of the Introduction section, authors should clearly discuss problematics, present the state of the art. The nature of the fertilizers used in this work should be given at this stage and clearly explain what is the novelty of their work, originality compared to previous works.

Authors: Add

Conclusion: Authors should avoid a "summary" and clearly explain what their work relevance for the progress in the field.

Authors: Add

It's regrettable to know that with such a work, performed with two plants, no pertaining photos that are self explaining are available.

Authors:  Do you have right; it is regrettable because we did not kept photos from experiment. We do not have photos to illustrate the differences between tomato or pepper according to the fertilizer type.
